# A Polyphasic Approach including Whole Genome Sequencing Reveals *Paecilomyces paravariotii* sp. nov. as a Cryptic Sister Species to *P. variotii*

**DOI:** 10.3390/jof9030285

**Published:** 2023-02-22

**Authors:** Andrew S. Urquhart, Alexander Idnurm

**Affiliations:** 1Commonwealth Scientific and Industrial Research Organisation, Saint Lucia, QLD 4067, Australia; 2Applied Biosciences, Macquarie University, Macquarie Park, NSW 2113, Australia; 3School of BioSciences, University of Melbourne, Parkville, VIC 3010, Australia

**Keywords:** *Eurotiales*, taxonomy, *Paecilomyces variotii*, horizontal gene transfer, viriditoxin

## Abstract

Whole genome sequencing is rapidly increasing phylogenetic resolution across many groups of fungi. To improve sequencing coverage in the genus *Paecilomyces* (*Eurotiales*), we report nine new *Paecilomyces* genomes representing five different species. Phylogenetic comparison between these genomes and those reported previously showed that *Paecilomyces paravariotii* is a distinct species from its close relative *P. variotii*. The independence of *P. paravariotii* is supported by analysis of overall gene identify (via BLAST), differences in secondary metabolism and an inability to form ascomata when paired with a fertile *P. variotii* strain of opposite mating type. Furthermore, whole genome sequencing resolves the *P. formosus* clade into three separate species, one of which lacked a valid name that is now provided.

## 1. Introduction

*Paecilomyces* Bainier is a cosmopolitan genus of 14 species in the order *Eurotiales* comprising 9 species as defined by the monograph on the genus by Samson et al. [1] (except that *P. saturatus* (Nakaz., Y. Takeda and Suematsu) Samson and Houbraken is invalid and correctly called *P. dactylethromorphus* Bat. and H. Maia [2]), *P. maximus* C. Ram and *P. lecythidis* C. Ram, which Samson et al. synonymised with *Paecilomyces formosus* (nom. inval.), which we now recognise as separate species, and 3 species that have been described subsequently (*P. clematidis* Spetik, Eichmeier, Gramaje and Berraf-Tebbal [3]; *P. penicilliformis* Jurjević and Hubka [4]; *P. tabacinus* Jurjević, Hubka, S.W. Peterson [5]). A number of entomopathogenic species previously included in the genus belong to the order *Hypocreales* [6]. Previously, several species have been known by teleomorph names in the genus *Byssochlamys* Westling (e.g., *P. variotii* Bainier = *B. spectabilis* (Udagawa and Shoji Suzuki) Houbraken and Samson); however, we follow Rossman et al. in using the earlier name *Paecilomyces* [7].

*Paecilomyces* species intersect with human activities in diverse ways. Several, notably *P. variotii*, have been reported as agents of human disease [7,8]. In other settings, *Paecilomyces* are important plant pathogens, for example, *P. lecythidis* has been identified as an agent of dieback in pistachio [9], and *P. niveus* (Stolk and Samson) as responsible for *Paecilomyces* rot in apples [10]. More positively, *P. variotii* has been increasingly investigated as a biocontrol agent for use in agriculture [11,12,13]. *Paecilomyces* species are also known for their heat tolerance, with the conidia of some strains representing the most heat-resistant fungal conidia (some conidia surviving over 22 min at 60 °C) [14]. Even more remarkably, their sexual spores can survive temperatures up to 85 °C for several minutes [15]. Hence, the high thermotolerance of these spore types means that *Paecilomyces* species are common agents of food spoilage in heat-treated food and drink [15].

*P. variotii* is an experimentally tractable organism in which transformation, homologous recombination for targeted gene replacements [16], CRISPR/Cas9 mutation [17,18], and genetic crosses are possible [15]. These tools have been used to make key insights into both secondary metabolite synthesis [19] and the discovery of host-beneficial transposons in eukaryotes [20]. Since the publication of three initial *Aspergillus* genomes in 2005 [21,22,23], whole genome sequencing coverage in the *Eurotiales* has been rapidly expanding; for example, more than 1,000 genomes of species in this order are accessible through NCBI, and *Paecilomyces* has been no exception with at least 14 assemblies now public. In addition, here we report nine new *Paecilomyces* genomes (Table 1). This increasing sequencing allows for better phylogenetic resolution in the genus by providing more phylogenetic markers. In this study, examination of multigene phylogenies indicates the separation of a laboratory strain, FRR 5287 (previously identified as *P. variotii* [20]), from other strains of the species. Mating experiments and examination of extrolite profiles supported the identification of a new species, described here as *P. paravariotii.* Additionally, we provide a taxonomical valid description for *Paecilomyces formosus*, which is separated from other species based on whole genome sequencing data.

## 2. Materials and Methods

### 2.1. Whole Genome Sequencing and Assembly

DNA was extracted from mycelia of *Paecilomyces* strains cultured in potato dextrose broth and then lyophilised, using a CTAB-based buffer and heating at 65 °C before treatment with chloroform and precipitation with isopropanol, as described previously [31]. Genomic DNA was sequenced using 150 nucleotide paired-end Illumina reads at either the Victorian Clinical Genetics Services or Australian Genome Research Facility. Reads were assembled using Velvet version 1.2.10 with a *k*-mer value of 100 [32]. Raw reads were deposited in BioProject PRJNA604095.

### 2.2. Properties of P. paravariotii in Culture

Colony morphology was observed on eight different media. Creatine sucrose agar (**CREA**) agar was prepared similarly to previously ([33]) and contained per litre 3 g creatine, 30 g sucrose, 1.3 g K_2_HPO_4_, 0.5 g MgSO_4_.·7H_2_O, 0.5 g KCl, 10 mg FeSO_4_·7H_2_O, 10 mg ZnSO_4_·7H_2_O, 50 mg bromocresol purple, 15 g agar, pH 8. Potato dextrose agar (**PDA**) was prepared from Difco potato dextrose agar powder. V8 juice agar (**V8**) agar contained 10% clarified Campbell’s V8 juice (adjusted to pH 6 with NaOH) and 2% agar. Yeast extract sucrose (**YES**) media contained 20 g yeast extract, 150 g sucrose, 0.5 g MgSO_4_·7H_2_O, 5 mg CuSO_4_, 10 mg ZnSO_4_·7H_2_O, 15 g agar [34]. Malt extract agar (**MEA**) media was prepared by supplementing Difco malt extract broth with 2% agar. **Minimal medium** contained per litre 10 g glucose, 6.0 g NaNO_3_, 0.52 g KCl, 0.52 g MgSO_4_·7H_2_O, 1.52 g KH_2_PO_4_ and 1 mL Hutner’s trace elements [35]. Yeast synthetic drop-out medium without uracil (**SD-ura**) was purchased from Merck. Czapek Yeast Autolysate (**CYA**) agar was prepared similar to [36] and contained per litre 3 g NaNO_3_, 5 g yeast extract, 30 g sucrose, 1.3 g K_2_HPO_4_, 0.5 g MgSO_4_·7H_2_O, 0.5 g KCl, 10 mg FeSO_4_·7H_2_O, 5 mg CuSO_4_, 10 mg ZnSO_4_·7H_2_O, 15 g agar.

A dried type specimen was deposited in the University of Melbourne Herbarium (MELU), and living cultures are available in the CSIRO Food Fungal Culture Collection (FRR) and Jena Microbial Resource Collection.

### 2.3. Phylogenetic Analyses

Nine-gene phylogeny: Single copy genes were chosen to construct a multilocus tree. Genes were chosen from the *P. variotii* CBS 101075 genome available from MycoCosm [16] and encoded actin-binding protein SLA2 (449012), chitin synthase activator (487455), methylenetetrahydrofolate reductase (260380), cell cycle control protein cwc22 (274517), CTD phosphatases (444213), domain of unknown function 726 (244097), mitotic check point protein bub2 (468177) and beta-tubulin (486644), where the numbers in parentheses represent JGI protein IDs. Nucleotide regions were aligned using MAFFT [37], concatenated and manually edited to remove poorly aligned regions (e.g., introns). The resulting alignment was 15,768 bp in length.

Four-gene phylogeny: To construct a tree including the closely related species *P. brunneolus* (N. Inagaki) Samson and Houbraken (for which no genome sequence is available), we utilised four markers, which are available both for the ex-type strains of *P. brunneolus* (CBS 370.70) and *P. variotii* (CBS 102.74), namely, calmodulin (EU037033.1/EU037038.1), beta-tubulin (EU037068.1/EU037073.1), actin (EU037016.1/EU037021.1) and RPB2 (MN969152.1/MN969153.1). As previously noted, nucleotide regions were aligned using MAFFT [30] and concatenated. The resulting alignment was 2,333 bp in length.

Phylogenetic trees were generated using two different approaches, a maximum likelihood approach using MEGA-X [31] and a Bayesian approach in MrBayes [38]. For MEGA, we first used the model finding tool (using the following settings: tree to use—automatic, complete deletion of gaps/missing data and no branch swap filter) to select the general time reversible (GTR) with gamma distribution and invariable sites as the best model base for the 9 gene phylogeny (log likelihood 140,968.324) and the Tamura 3-parameter model with gamma distribution of the 4-gene tree. Trees were generated using the chosen models with settings for 9 gamma categories, default tree inference options and 100 bootstrap replicates. MrBayes was implemented within Geneious Prime version 11.0.4 with the following settings: substitution model GTR; gamma variation with 9 categories; chain length 1,100,000; 4 heated chains; heated chain temp 0.2; subsampling frequency 200; burn-in length 100,000, random seed 22,494; unconstrained branch lengths GammaDir (1,0.1,1,1).

### 2.4. Metabolite Analysis Using High-Performance Liquid Chromatography (HPLC) and Mass Spectrometry

Strains were grown in 25 mL stationary potato dextrose broth cultures for five days. Mycelia were filtered out through miracloth, and the metabolites were extracted from the spent culture filtrate with an equal volume of ethyl acetate, dried under nitrogen, and then run on a Waters ACQUITY HPLC-PDA–QDa system. HPLC separation was achieved on a C18 column (2.7 µm, 2.1 × 100 mm Cortecs 186007367) using a linear solvent gradient from 5% to 100% acetonitrile over 10 min. Solvents were supplemented with 0.1% formic acid. UV absorbance was measured at 254 nm. The QDa detector was used to scan in negative mode from 150–1000 Daltons with a cone voltage of 10 V, a sampling rate of 4.2 points/second and a capillary voltage of 0.8 kV.

### 2.5. Sexual Crosses

Crosses were conducted as described previously for *P. variotii* [15,16]. Briefly, strains of opposite mating types were inoculated as parallel streaks approximately 20 mm apart on 90 mm PDA plates and incubated at 30 °C for up to 6 weeks, after which time plates were examined for ascomata. Additionally, ascospore/conidia mixtures were picked from the plates (by scrapping a small amount of material, equivalent to approximately 5 µL volume, with a pipette tip), suspended in water, and heat treated at 80 °C for 10 min on a dry heat block, after which only the ascospores remain viable [1]. The heat-treated spore mixtures (containing live ascospores, if present, and dead asexual spores) were plated onto V8 agar, and colonies were observed after 48 h at 30 °C.

### 2.6. Transformation

Strain FRR 5287 was transformed with a transfer-DNA delivery by *Agrobacterium tumefaciens* using the same method and plasmid that includes DNA to express histone 2b fused to CFP and hygromycin resistance as reported for *P. variotii* [16].

### 2.7. Microscopy

Micromorphology was examined using a standard slide culture procedure on PDA after 48 h at 30 °C.

For light and fluorescence microscopy, either a Leica DM6000 or a Zeiss Axio Imager M2 was used. For scanning electron microscopy (SEM), fungi were grown on nitrocellulose membranes placed on PDA, then transferred to SEM stubs, sputter coated with gold using a Quorum 150T ES plus machine and visualised with a Hitachi TM4000 Plus scanning electron microscope.

## 3. Results and Discussion

### 3.1. ‘Paecilomyces variotii’ Strain FRR 5287 Is Phylogenetically Separated from Previously Described Paecilomyces Species

Previous research to explore the distribution of a transposable element in *P. variotii* and related *Eurotiales* included a number of *Paecilomyces* strains. Due to ambiguity in assignment of strains to different *Paecilomyces* species, we also sequenced a standard phylogenetic marker and noticed that *P. variotii* strain FRR 5287 appeared outside or basal to other *P. variotii* strains [20]. Exploring this strain further using whole genome sequencing information suggested it represents a distinct species.

Phylogenetic trees constructed from nine single copy genes (extracted from whole genome sequencing supplemental Table 1) were highly concordant with a tree generated from a concatenated alignment (Figure 1A). For each region tested, there was clear separation between FRR 5287 and other *P. variotii* isolates. Below the species level in *P. variotii*, individual genes were highly discordant, a finding consistent with naturally occurring sexual reproduction (as previously reported [15]).

The only previously described *Paecilomyces* species closely related to *P. variotii* and for which no genome sequencing information is available is *P. brunneolus* [1]. We therefore generated a phylogenetic tree including *P. brunneolus* based on four markers (actin, calmodulin, beta-tubulin, RPB2). Each of these markers demonstrated that *P. paravariotii* FRR 5287 and *P. brunneolus* are not co-specific (Figure 1B). Sequences for each of these for markers from *P. paravariotii* have been deposited in GenBank (OP985492–OP985495). BLAST searches against the NCBI database revealed that *P.* “*variotii*” CCF 6349 shows a 100% identity match to *P. paravariotii* FRR 5287 in the beta-tubulin and calmodulin regions (LR778164.1 and LR778166.1 [4]), suggesting that the strain represents an additional isolate of *P. paravariotii.*

### 3.2. Genome-Wide BLAST Comparisons Confirm Divergence between P. variotii and P. paravariotii and Reveal a Previously Unrecognised Horizontal Gene Transfer (HGT) Event

Given the separation between *P. paravariotii* and *P. variotii* in the phylogenetic analysis, we decided to assess the level of divergence between these species on a genome-wide scale. To do this, we took the gene annotations for *P. variotii* CBS 144490 (8168 genes), used this with BLAST against the genome of the species to be compared against, and then graphed the top hit for each gene on a plot of identity vs. length. As expected, this revealed greater divergence between *P. variotii* CBS 144490 and *P. paravariotii* FRR 5287 than between *P. variotii* CBS 144490 and other *P. variotii* strains (Figure 2).

We previously demonstrated the presence of a large transposable element called *hephaestus* in strains assigned as *P. variotii* and, in one strain of *P. lecythidis* [20,24], an element that increases resistance to at least five toxic metal ions. Subsequently, we have shown evidence to support HGT of *hephaestus* between *P. variotii* and *P. lecythidis* [24]. In the *P. variotii* CBS 144490 and *P. paravariotii* FRR 5287 comparison, a set of genes (highlighted in blue) were 100% identical in nucleotide sequence between *P. variotii* and *P. paravariotii*. This striking conservation is compared to the rest of the genes in the genome, which have identities typically lower than 98%. These genes belong to *hephaestus*, suggesting that this transposon has moved horizontally, i.e., across species, into *P. paravariotii*. This finding serves as a reminder of the power of correct taxonomic delimitation of laboratory strains, which, in this case, has revealed a previously unrecognised HGT event.

### 3.3. FRR 5287 Is Unable to Form Ascospores with a Fertile P. variotii Strain of Opposite Mating Type

Given that the phylogenetic evidence indicated an absence of recombination between FRR 5287 and *P. variotii* we attempted to cross FRR 5287 to *P. variotii*. Examination of the genome sequence of FRR 5287 revealed a candidate mating type (*MAT*) locus, with the presence of a *mat1-2* allele flanked by the conserved *sla2* and *apn2* genes and consistent with a predicted heterothallic mode of sexual reproduction (Figure 3A). We attempted to cross FRR 5287 to strain CBS 101075 (*mat1-1*), which has been crossed in previous studies [15,16]. *P. variotii* CBS 144490 (*mat1-2*) was used as a positive control. After 4 weeks incubation, abundant ascomata were observed in the control cross, but none were observed in the *P. variotii* CBS 144490 × FRR 5287 cross across six replicate plates (Figure 3B). Consistent with the lack of visible ascomata, the cross to CBS 144490 yielded heat-resistant spores (i.e., ascospores), while the cross to FRR 5287 did not (Figure 3C). While the impossibility of mating cannot be inferred from a single pair of strains given that even within species not every strain pair is fertile, the lack of mating observed is consistent with the genome analyses, which indicates an absence of recombination (Figure 1 and Figure 2). If additional *P. paravariotii* strains are isolated, it will be valuable to test for mating both to *P. variotii* and intraspecifically.

### 3.4. Strain FRR 5287 Is Chemically Distinguished from P. variotii Due to Genetic Differences in Viriditoxin Biosynthesis

A feature of *P. variotii* is the production of the secondary metabolite viriditoxin **1** [1]. A nine-gene cluster is responsible for viriditoxin biosynthesis [19]. Differences in secondary metabolite production have previously been explored as taxonomic traits in the *Eurotiales*, including in *Aspergillus* [39] and *Paecilomyces* [1]. To determine whether FRR 5287 was expected to produce viriditoxin, we examined the genome for the presence of homologs of these genes. Indeed, a highly similar gene cluster is present within the genome. However, relative to that of sequenced *P. variotii* strains, it contains disruptions in two genes (Figure 4A). The first is the insertion of ~5.6 kb of AT-rich sequence into *vdtX* (a gene of unknown function, not required for viriditoxin production). The second is *vdtE* that in FRR 5287 contains both stop codons and frameshift mutation. The gene encodes a Baeyer–Villiger monooxygenase required to add two oxygens in the production of viriditoxin (Figure 4B) [19]. A *vdtE* deletion mutant of *P. variotii* thus produces compound **1** in place of viriditoxin (Figure 4B; [19]).

From the results of analysing putative gene functions, we hypothesised that FRR 5287 might produce compound **1** rather than viriditoxin. HPLC-UV-MS of wild type *P. variotii* showed a maximum peak at around 8.32 min of the UV trace, and the corresponding mass spectrum contained an ion at 661.17 (+/−0.2) consistent with the [M-H]^−^ ion of viriditoxin (monoisotopic mass 662.16) (Figure 4C,D). In FRR 5287, this peak was not observed (Figure 4C). The maximum peak in this isolate is around 7.84 min, and the mass spectrum shows a dominant ion of 629.09 (+/−0.2) consistent with the [M-H]^−^ ion of **1** (monoisotopic mass 630.17) (Figure 4E). These findings confirm our prediction that FRR 5287 would be unable to produce viriditoxin but instead a putative precursor molecule. Thus, *P. paravariotii* FRR 5287 can be chemically distinguished from other *P. variotii* strains. If more *P. paravariotii* isolates are discovered and characterised, it will be interesting to see if the production of compound **2** is a consistent and potentially diagnostic trait of *P. paravariotii*.

### 3.5. Features and Taxonomic Description of Paecilomyces Paravariotii

In addition to the difference identified in secondary metabolite production and underlying genetic basis, other phenotypic differences between strain FRR 5287 and the best characterised strain of *P. variotii*, CBS 144490, were explored. This included growth on different media, micromorphology, and genetic transformation with a reporter gene that enables a measurement of nucleus content (Figure 5). The following illustration, descriptions and notes are based on these findings.

Taxonomy.

***Paecilomyces paravariotii*** Urquhart, **sp. nov.**

MycoBank: MB 846976

**Holotype:** Isolated from unknown substrate, presumably USA 1976 or earlier (type MELUF155137a, a dried specimen of colony on filter paper). **Isotype**: Ex-type culture: FRR 5287.

Etymology: Greek, *para-* meaning similar to + *-variotii*, referring to the fungus *Paecilomyces variotii*.

Colony morphology is highly variable depending on media, reaching in 3 days at 30 °C a diameter of 16 mm on CREA, 14 mm on CYA, 38 mm on MEA, 17 mm on minimal media, 24 mm on PDA, 26 mm on V8 and 22 mm on YES. It had a similar growth rate at 37 °C on PDA. Conidiophores macronematous, mononematous, cylindrical, multibranched, somewhat verticillate branched towards the apex, smooth, hyaline 9.7–19.3 μm in length and 1–4 μm at the base. Conidiogenous cells monophialidic, discrete, subulate, determinate, arranged in verticils, smooth-walled, hyaline. Conidia basocatenate, ellipsoidal to subfusiform, unicellular, smooth, yellow–brown, 3.1–5.0 × 2.2–3.3 μm. Ascomata not observed in pure culture, presumably heterothallic. Poor growth and no acid production on CREA. No viriditoxin synthesis. Conidia are predominantly uninucleate.

Notes: Isolate obtained as FRR 5287 from the CSIRO FRR culture collection. The FRR culture collection obtained the isolate from the DSTO culture collection (culture number 1357) who, in turn, obtained the isolate from Proctor and Gamble (Cincinnati, OH, USA) via the University of Wisconsin. Unfortunately, the substrate from which the isolate was obtained is not known to us. *P. paravariotii* is morphologically alike to its sister species *P. variotii*, both of which can be distinguished from the *P. formosus/maximus/lecythidis* clade by the lack of acid production on CREA (Appendix A). *P. paravariotii* can be distinguished from *P. variotii* based on an absence of viriditoxin and the production of conidia that are predominantly uninucleate.

***Paecilomyces formosus*** Urquhart, **sp. nov.**

MycoBank: MB 846977

= *Monilia formosa* Sakag., May. Inoue and Tada, Zentralbl. Bakteriol., 2. Abt. 100: 302. 1939. (nom. inval.) [MB 252219]

= *Paecilomyces formosus* (Sakag., May. Inoue and Tada) Houbraken and Samson (nom. inval.) [MB 512562]

Holotype: isolated from a botanical specimen preserved in a dilute formaldehyde solution, Taiwan, 1939 (CBS 990.73B, culture preserved in a metabolically inactive state).

Etymology: named as per the invalidly described *Paecilomyces formosus*.

Description as per that given for “*Paecilomyces formosus*” by R.A. Samson, J. Houbraken, J. Varga, J.C. Frisvad Persoonia 22, 2009: on page 21 and continued page 24.

Notes: We provide a valid name for the taxon introduced invalidly as *Monilia formosa* Sakag., May. Inoue and Tada. Samson et al. (2009) attempted to introduce the new combination “*Paecilomyces formosus* (Sakag., May. Inoue and Tada) Houbraken and Samson”, but this name is invalid, being based on the earlier invalid name, and it is also illegitimate, because Samson et al. (2009) included in synonymy two earlier names (*P. maximus* C. Ram and *P. lecythidis* C. Ram,) one of which should have been adopted. Following Art. 46.4 of the International Code of Nomenclature, we do not ascribe the name *P. formosus* to “Sakag., May. Inoue and Tada”, as these authors introduced the epithet *formosus* in *Monilia*, which is a different genus, to the current placement in *Paecilomyces*. *P. formosus* is clearly genetically distinct from other *Paecilomyces* species, including the most closely related species *P. maximus* and *P. lecythidis*. We previously found that in *P. variotii*, the majority of conidia are multinucleate whereas in *P. paravariotii* most conidia contain only one nucleus—this is thus a potentially informative taxonomic character that should be explored more broadly across the genus including a greater number of strains.

## 4. Conclusions

Cryptic species are commonly found in fungal lineages and can be distinguished by a number of methods [40]. In this study, exploring the nucleotide divergence of a fungal strain hinted to the presence of an unrecognised species, which was established based on phylogenetic, phenotypic and mating incompatibility with its closest relatives. A limitation of this study is that the phenotypic differences between *P. paravariotii* and *P. variotii* are based around a single strain of *P. paravariotii*, and hence, how consistent and distinguishing such characteristics are remains to be further established. Given that at least one additional putative strain of *P. paravariotii*, CFF 6349 [4], is present as based on sequences in GenBank, examination of additional strains may be possible.

## Figures and Tables

**Figure 1 jof-09-00285-f001:**
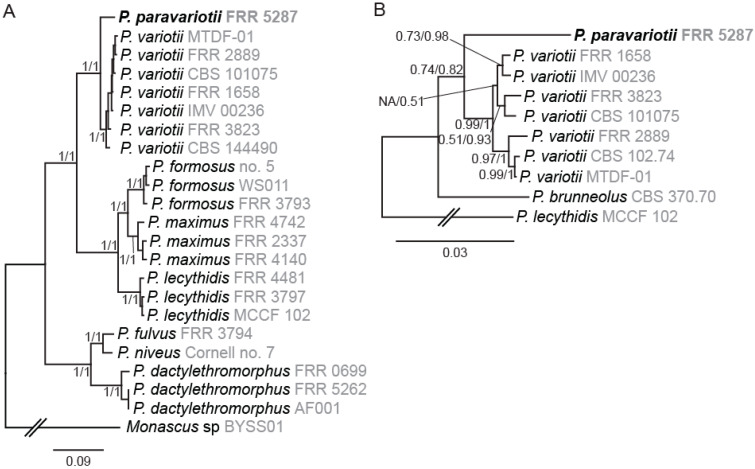
Phylogenetic analysis supports the separation of *Paecilomyces* strain FRR 5287 into a new species. (**A**) Nine-gene tree showing relationships among genome sequenced *Paecilomyces* strains. (**B**) Four-gene tree showing relationships among the *P. variotii*/*paravariotii*/*brunneolus* clade. Trees shown were generated in MrBayes. Branch supports indicate bootstraps from maximum likelihood analysis in MEGA-X (first number) and Bayesian probabilities from MrBayes (second number).

**Figure 2 jof-09-00285-f002:**
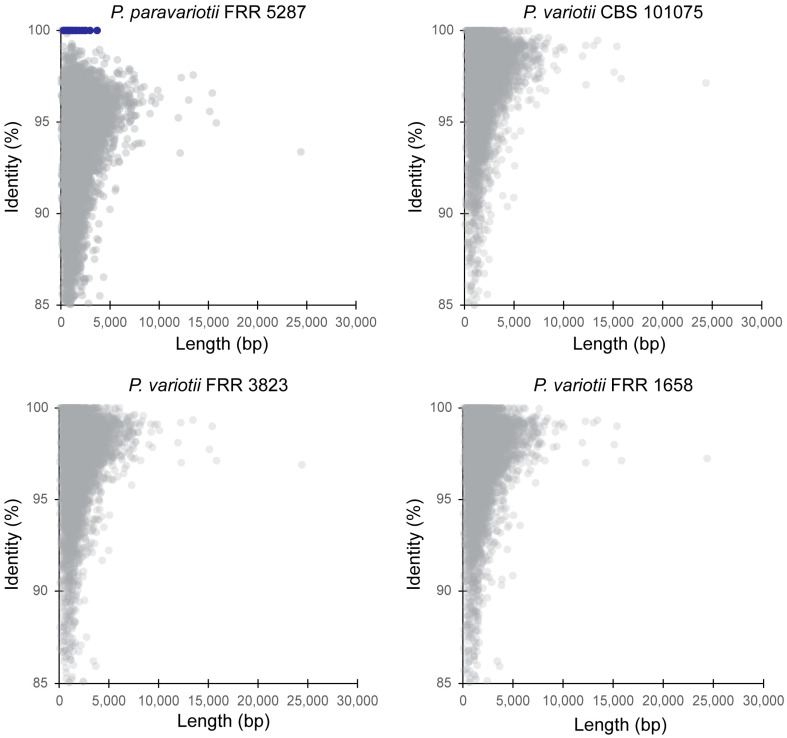
Genome-wide assessment of divergence between *P. variotii* and *P. paravariotii* using a BLAST all-vs.-all approach. Each comparison is to the *P. variotii* CBS 144490 gene regions. Blue colour indicates genes contained within the *hephaestus* transposable element.

**Figure 3 jof-09-00285-f003:**
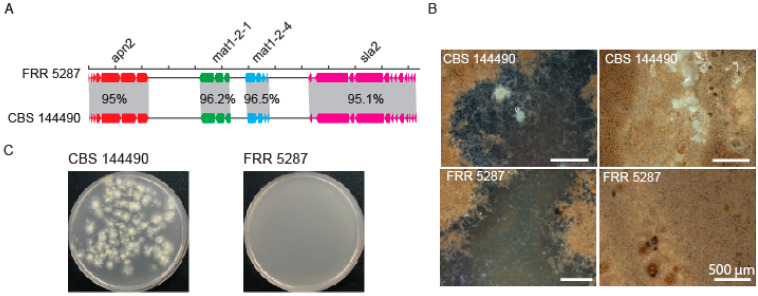
Strain FRR 5287 is reproductively isolated from *P. variotii*. (**A**) Diagram of mating type loci. (**B**) Ascomata production of mating plates. (**C**) Germination of ascospores following heat treatment of mating material; only the positive control yielded colonies.

**Figure 4 jof-09-00285-f004:**
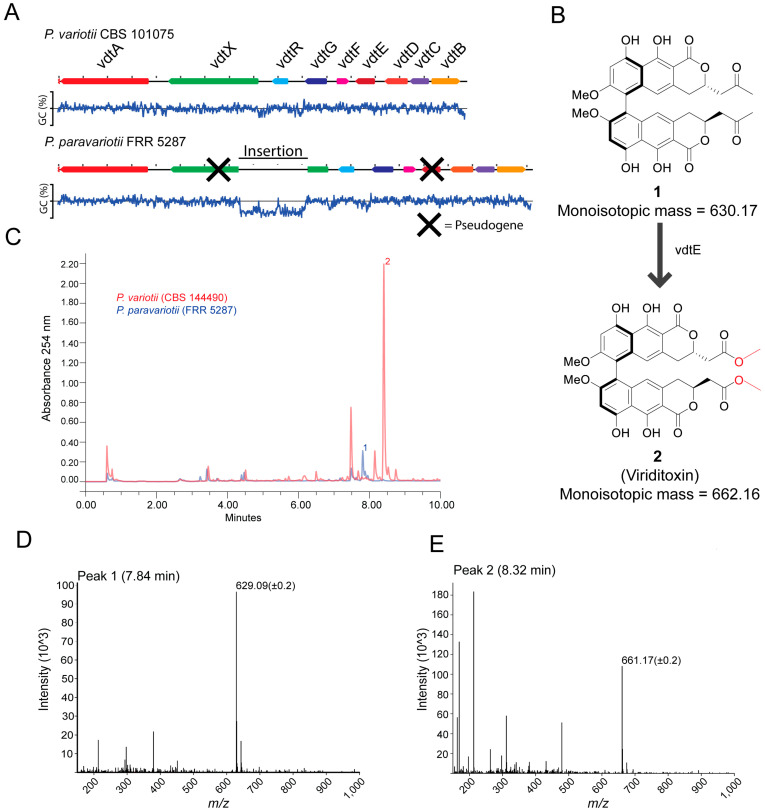
*P. paravariotii* strain FRR 5287 is unable to synthesis mature viriditoxin. (**A**) FRR 5287 has a mutated viriditoxin biosynthesis cluster in which *vdtX* and *vdtE* are pseudogenes. The % GC content across the DNA is graphed below the DNA region. (**B**) *vdtE* is responsible for the Baeyer–Villiger oxidation in viriditoxin biosynthesis; its absence leads to the production of compound **1** rather than viriditoxin **2**. (**C**) HPLC-MS data showed the production of **1** in FRR 5287. The mass spectra for (**D**) compound **1** and (**E**) compound **2**.

**Figure 5 jof-09-00285-f005:**
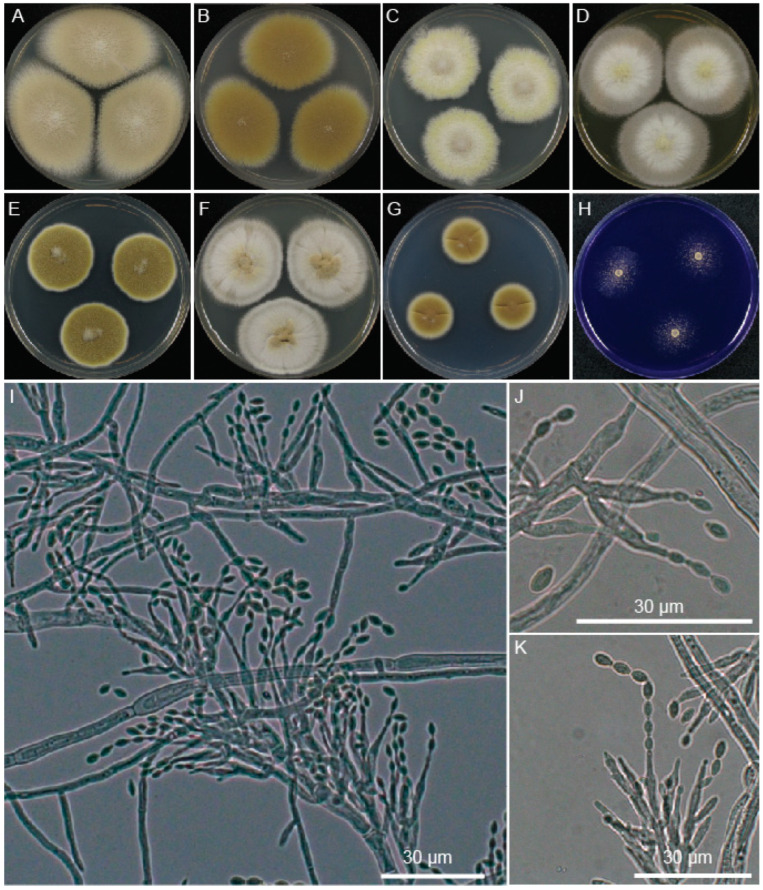
Morphology of *P. paravariotii* FRR 5287. Five-day-old colonies grown at 30 °C on 90 mm plates of (**A**) MEA, (**B**) V8, (**C**) SD-ura, (**D**) YES, (**E**) PDA, (**F**) CYA, (**G**) minimal, and (**H**) CREA. (**I**–**M**) Conidiophores. (**N**) Conidia overlayed with histone-H2B CFP fluorescence.

**Table 1 jof-09-00285-t001:** Sequenced *Paecilomyces* genomes.

GenBank Accession	GenBank Name	Corrected Name	Citation
JAPVCG000000000	*P. fulvus* FRR 3794	-	This study
JAPVCF000000000	*P. dactylethromorphus* FRR 0699	-	This study
JAPVCE000000000	*P. dactylethromorphus* FRR 5262	-	This study
JAPVCD000000000	*P. lecythidis* FRR 3797	-	This study
JAPVCC000000000	*P. lecythidis* FRR 4481	-	This study
JAPVCB000000000	*P. formosus* FRR 3793	-	This study
JAPVCA000000000	*P. maximus* FRR 4742	-	This study
JAPVBZ000000000	*P. maximus* FRR 2337	-	This study
JAPVBY000000000	*P. maximus* FRR 4140	-	This study
JANCMQ000000000	*P. variotii* FRR 1658	-	[24]
JANCMP000000000	*P. variotii* FRR 2889	-	[24]
JANCMO000000000	*P. variotii* FRR 3823	-	[24]
JANCMN000000000	*P. variotii* FRR 5287	*P. paravariotii* FRR 5287	[16]
RHLL00000000.1	*P. variotii* CBS 144490	-	[16]
RCNU00000000.1	*P. variotii* CBS 101075	-	[16]
MSJH00000000.2	*Byssochlamys* sp. IMV 00236	*P. variotii* IMV 00236	[25] (*Cladosporium cladosporioides* IMV 00236 in publication)
QBDR00000000.1	Thermoascaceae sp. COH1141	*P. lecythidis* COH1141	[26]
JAGJCD000000000.1	*Aspergillus* sp. MCCF 102	*P. lecythidis* MCCF 102	
BAUL01000000.1	*Byssochlamys spectabilis* No. 5	*P. formosus* no. 5	[27]
JACXGS000000000.1	*Paecilomyces variotii* WS011	*P. formosus* WS011	
PNEM00000000.1	*Byssochlamys* sp. AF001	*P. dactylethromorphus* AF001	[28]
QEIL00000000.1	*Paecilomyces niveus* Cornell no. 7	-	[29]
RCHW01000000.1	*Paecilomyces variotii* MTDF-01	-	[30]

## Data Availability

All data are presented in the manuscript, associated supplemental data or deposited to the GenBank database.

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
