# Peer review of "A Polyphasic Approach including Whole Genome Sequencing Reveals Paecilomyces paravariotii sp. nov. as a Cryptic Sister Species to P. variotii"

_jof, 2023, doi:10.3390/jof9030285_

Round 1
Reviewer 1 Report
This paper describe and illustrate a crytic new species based on morphological and genomic analyses. As discussed by the authors, more strains of the new species should be studied to strength the separation.
My main concern is about the Paecilomyces formosus sp. nov. Urquhart. Since the epithet "formosus" was already used as "Paecilomyces formosus Sakag., May. Inoue & Tada ex Houbraken & Samson", the authors can not use the epithet when describing it as a new species. Please consult the codes or senior taxonomists for the nomenclature problem of the species. In addition, I suggest to include this part of contents in the title of the paper.
Author Response
As attached

Reviewer 2 Report
I think the manuscript is written well. Experimental design and results are clear. However, some revisions are required for publication.
There are few comments in morphological descriptions of fungal species.
1. The full names should be defined first such as creatine sucrose agar (CREA), malt extract agar (MEA).
2. The description of new species (Paecilomyces paravariotii) is too simplified. For example, no growth data on various agar media MEA, YES, CYA, CREA are mentioned in the description section. Please check the ref. Samson et al. 2009; Persoonia 22, 2009: 14–27. Size of conidia should be inserted in the text.
3. Method for ascospore germination (rate) test in liquid or solid media should be described in Method section in detail. In Fig. 3 C, authors need to describe experiment conditions in detail.
Author Response
as attached

Reviewer 3 Report
This paper entitled"Whole genome sequencing reveals Paecilomyces paravariotii sp. nov. as a cryptic sister species to P. variotii" is appeared to be a nice piece of work and will provide more information and reference for future study. However, I think this need major revised in the present form. The main problem are as follows:
1. As a taxonomy article, I would like suggest the author to add named person, when the species name appears for the first time.
2. I would like to suggest the author to add more morphylogical characteristics of Paecilomyces paravariotii. There is only the characteristics of conidiophores, which could not enough for latter research and identification. The size of phialide and conidia were important for this fungi.
3. I would like to suggest the author to reconsider the title of the paper. The multi-gene phylogenetic analysis, mating type and metabolite analysis were contained in the paper. In my opinion, I would like to focus on the whole genomic sequence to establish of the new species.
4. I would like to suggest the author to reanalysis the phylogenetic tree by MrBayes, RAxML or IQ-TREE, which could screening suitable model for multi-gene. Besides, the detailed steps need to be listed.
Author Response
as attached

Round 2
Reviewer 2 Report
This manuscript has been improved through a revision process. However, I have only one comment. Figure S1 should be mentioned in the text in page 11 line 21.
It is now acceptable.
Author Response
We have added the required reference to the SI figure. We thank the reviewer very much for spotting this omission.